

# BiSCoT: improving large eukaryotic genome assemblies with optical maps

Benjamin Istace, Caroline Belser and Jean-Marc Aury

Génomique Métabolique, Genoscope, Institut François Jacob, CEA, CNRS, Univ Evry, Université Paris-Saclay, Evry, France

## ABSTRACT

**Motivation**. Long read sequencing and Bionano Genomics optical maps are two techniques that, when used together, make it possible to reconstruct entire chromosome or chromosome arms structure. However, the existing tools are often too conservative and organization of contigs into scaffolds is not always optimal.

**Results**. We developed BiSCoT (Bionano SCaffolding COrrection Tool), a tool that post-processes files generated during a Bionano scaffolding in order to produce an assembly of greater contiguity and quality. BiSCoT was tested on a human genome and four publicly available plant genomes sequenced with Nanopore long reads and improved significantly the contiguity and quality of the assemblies. BiSCoT generates a fasta file of the assembly as well as an AGP file which describes the new organization of the input assembly.

**Availability**. BiSCoT and improved assemblies are freely available on GitHub at http://www.genoscope.cns.fr/biscot and Pypi at https://pypi.org/project/biscot/.

## INTRODUCTION

Assembling large and repetitive genomes, such as plant genomes, is a challenging field in bioinformatics. The appearance of short reads technologies several years ago improved considerably the number of genomes publicly available. However, a high proportion of them are still fragmented and few represent the chromosome organization of the genome. Recently, long reads sequencing techniques, like Oxford Nanopore Technologies and Pacific Biosciences, were introduced to improve the contiguity of assemblies, by sequencing DNA molecules that can range from a few kilobases to more than a megabase in size (*Istace et al., 2017*; *Schmidt et al., 2017*; *Kim et al., 2019*; *Shafin et al., 2019*). Nevertheless and even if the assemblies were greatly improved, the chromosome-level organization of the sequenced genome cannot be deciphered in a majority of cases. In 2017, Bionano Genomics launched its Saphyr system which was able to generate optical maps of a genome, by using the distribution of enzymatic labelling sites. These maps were used to orient and order contigs into scaffolds but the real improvement came in 2018, when Bionano Genomics introduced their Direct Label and Stain (DLS) technology that was able to produce genome maps at the chromosome-level with a N50 several times higher than previously (*Belser et al., 2018*; *Formenti et al., 2018*; *Hu et al., 2019*).

Corresponding author
Benjamin Istace,
bistace@genoscope.cns.fr

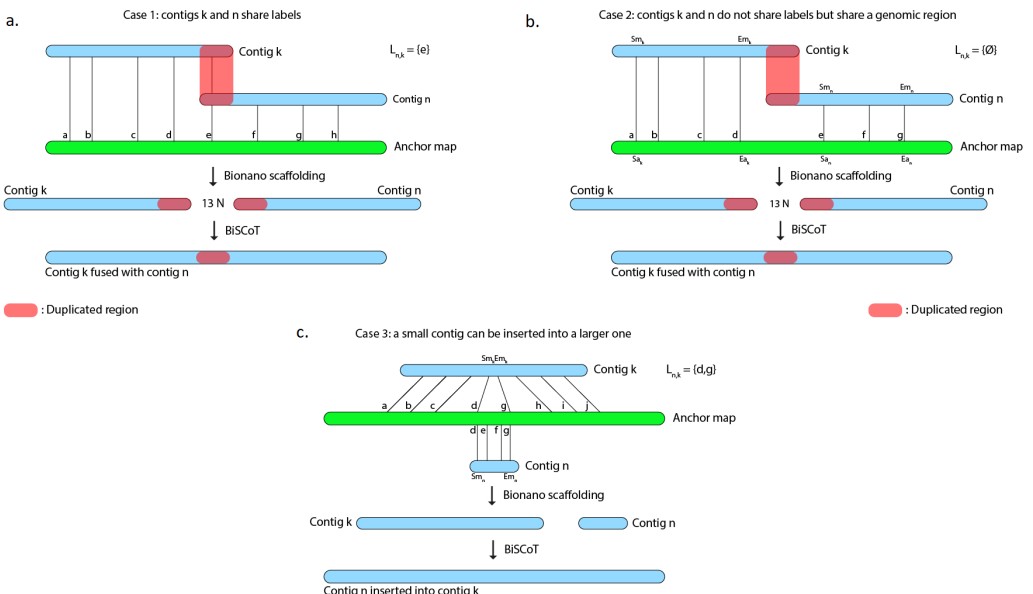

**Figure 1 The Bionano scaffolding tool does not merge contigs even if they share labels.** Instead, it inserts 13 N's gap between contigs, thus artificially duplicating the shared region. (A) BiSCoT merges contigs that share enzymatic labelling sites. (B) If contigs do not share labels but share a genomic region, BiSCoT attempts to merge them by aligning the borders of the contigs. (C) The Bionano scaffolding tool does not handle cases where contigs can be inserted into others. BiSCoT attempts to merge the inserted map with the one containing it if they share labels.

However, scaffolds generated with the tool provided by Bionano Genomics do not reach optimal contiguity. Indeed, when two contigs $C_1$ and $C_2$ are found to share labels, one could expect that the tool would merge the two sequences at the shared site. Instead, the software chooses a conservative approach and outputs the sequence of $C_1$ followed by a 13-Ns gap and then the $C_2$ sequence, thus duplicating the region that is shared by the two contigs (Fig. 1A and 1B) and in numerous cases, these duplicated regions could reach several kilobases. As an example, on the human genome we used to evaluate BiSCoT (see 'Results'), we could detect 515 of those regions, affecting 16 genes and corresponding to around 24.5 Mb of duplicated sequences, the longest being 237 kb in size. These duplicated regions affect the contiguity and have to be corrected as they can be problematic for downstream analyses, like copy number variation studies. They originate from overlaps that are not fused in the input assembly and usually correspond to allelic duplications. In addition, contigs can sometimes be inserted into other contigs, these cases are not handled by the Bionano scaffolding tool that discards the inserted contigs (Fig. 1C).

We developed BiSCoT, a python script that examinates data generated during a previous Bionano scaffolding and merges contigs separated by a 13-Ns gap if needed. BiSCoT also re-evaluates gap sizes and searches for an alignment between two contigs if the gap size is inferior to 1,000 nucleotides. BiSCoT is therefore not a traditional scaffolder since it can only be used to improve an existing scaffolding, based on an optical map.

## METHODS

### Mandatory files loading

During the scaffolding, the Bionano scaffolder generates a visual representation of the hybrid scaffolds that is called an 'anchor'. It also generates one 'key' file, which describes the mapping between map identifiers and contig names, several CMAP files, which contain the position of enzymatic labelling sites on contig maps and on the anchor, and a XMAP file, that describes the alignment between a contig map and an anchor. BiSCoT first loads the contigs into memory based on the key file. Then, the anchor CMAP file and contig CMAP files are loaded into memory. Finally, the XMAP file is parsed and loaded.

### Scaffolding

Alignments of contigs onto anchors contained in the XMAP file are first sorted by their starting position on the anchor. Then, alignments on one anchor are parsed by pairs of adjacent contigs, i.e alignment of contig $C_k$ is examined at the same time as contig $C_n$, with $C_k$ aligned before $C_n$ on the anchor. Aligned anchor labels are extracted from these alignments and a list of shared labels $L_{n,k}$ is built. For the following cases, we suppose $C_k$ and $C_n$ to be aligned on the forward strand (Fig. 1).

#### *Case 1: contig maps share at least one anchor label*

The last label $l$ from $L_{n,k}$ is extracted and the position $P_l$ of $l$ on both contigs $C_k$ and $C_n$ is recovered from the CMAP files. In the resulting scaffold, the sequence of $C_k$ will be included up to the $P_l$ position and the sequence of $C_n$ will be included from the $P_l$ position. In this case, the gap is removed, both contigs $C_k$ and $C_n$ are fused and BiSCoT generates a single contig instead of two contigs initially separated by a gap in the input assembly.

#### *Case 2: contig maps do not share anchor labels*

Let $Size_k$ be the size of the contig $C_k$, $Sm_k$ and $Em_k$ the start and end of an alignment on a contig map and $Sa_k$ and $Ea_k$ the corresponding coordinates on the anchor. The number $n$ of bases between the last aligned label of $C_k$ and the first aligned label of $C_n$ is then:

$$n = Sa_n - Ea_k \tag{1}$$

We then have to subtract the part $d_k$ of $C_k$ after the last aligned label of $C_k$ and the part $d_n$ of $C_n$ before the first aligned label of $C_n$:

$$d_k = Size_k - Em_k \tag{2}$$

$$d_n = Sm_n \tag{3}$$

Finally, we can compute the gap size $g$ with:

$$g = n - d_k - d_n \tag{4}$$

If $g \leq 1000$, a BLAT (*Kent, 2002*) alignment of the last 30 kb of $C_k$ is launched against the first 30kb of $C_n$. If an alignment is found and if its score is higher than 5,000, $C_k$ and $C_n$ are merged at the starting position of the alignment and, as in case1, BiSCoT generates a single contig instead of two contigs initially separated by a gap in the input assembly. Otherwise, a number $g$ of Ns is inserted between $C_k$ and $C_n$.

***Case 3: insertion of small contigs***
Let $Sm_k$ and $Em_k$ the start and end of an alignment on a contig map. If $[Sm_n, Em_n] \subset [Sm_k, Em_k]$, then the left-most shared label identifier $l_l$ and right-most shared label identifier $l_r$ are extracted. If $C_n$ has more of its labels mapped in this region than $C_k$, the sequence of $C_n$ will be inserted between $l_l$ and $l_r$ in the scaffolds. Otherwise, the sequence of $C_k$ remains unchanged and $C_n$ will be included as a singleton sequence in the scaffolds file.

Finally, if an Illumina polishing step was done before or after Bionano scaffolding, we recommend doing one additional round of polishing using Illumina reads after BiSCoT has been applied. Indeed, short reads tend to be aligned only against one copy of the duplicated regions, leaving the other copy unpolished.

## RESULTS AND DISCUSSIONS

### Validation on simulated data

In order to simulate a genome assembly, we downloaded the chromosome 1 of the GRCh38.p12 human reference genome and fragmented it to create contigs. We generated 120 contigs with an N50 size of 2.4 Mb and a cumulative size of 231 Mb. Contigs were generated with either overlaps or gaps between them. We introduced 50 gaps with a mean length of 50 kb, the smallest being 3.4kbp long and the largest 99.6 kb long, and 50 overlaps with a mean size of 44kb, the smallest being 278b long and the largest 98.6 kb long. We also generated five contigs, with an N50 of 254 kb, that were subsequences of larger contigs, to simulate contained contigs.

Then, we used these contigs and Bionano DLE and BspQI optical maps available on the Bionano Genomics website as input to the Bionano scaffolder. We gave the results of this scaffolding to BiSCoT and aligned all assemblies to the chromosome 1 reference using Quast (*Gurevich et al., 2013*, v5.0.2).

BiSCoT was able to resolve 39 overlaps out of the 50 we introduced (Table S1), 31 using shared labels and 8 using a Blat alignment. The 11 remaining overlaps could not be resolved due to contigs not sharing enough labels or the overlap being too small to produce an alignment of sufficient confidence. BiSCoT was also able to integrate all contained contigs back to their original place in the assembly. Furthermore, BiSCoT did not close any of the real gaps introduced during the assembly generation.

Regarding assembly metrics (Table S2), The N50 decreased by 1.4% in scaffolds and increased by 22% in contigs. The number of Ns in scaffolds decreased from 20.7Mb to 20.4Mb. Moreover, the number of misassemblies decreased by 68% after applying BiSCoT and the duplication ratio estimated by Quast decreased from 1.026 in Bionano scaffolds to 1.021 in BiSCoT scaffolds.

In order to estimate the accuracy of gap sizes, we compared the gap sizes we introduced in the input assembly to the ones that were estimated using optical maps (Fig. S1). We found that estimated gap sizes were very close to the reality, with a mean scaled absolute error of 0.8%.

**Table 1   Metrics of the NA12878 scaffolds and contigs before or after BiSCoT treatment.** Bold formatting indicates the best scoring assembly among contigs.

| | Nanopore contigs | Bionano | | BiSCoT | |
|---|---|---|---|---|---|
| | | **Contigs** | **Scaffolds** | **Contigs** | **Scaffolds** |
| Cumulative size | 2,818,937,673 | 2,818,997,568 | 2,878,230,106 | 2,810,480,725 | 2,868,077,379 |
| N50 | 11,821,944 | 10,566,783 | 86,858,024 | **12,894,141** | 86,833,728 |
| L50 | 67 | 71 | 14 | **64** | 14 |
| N90 | 2,143,851 | 1,863,173 | 26,054,782 | **2,321,940** | 26,037,000 |
| L90 | 280 | 301 | 36 | **254** | 36 |
| auN[a] | 15,164,719 | 14,547,428 | 82,760,251 | **15,977,835** | 82,474,548 |
| # Ns | 0 | 0 | 59,232,538 | 0 | 57,596,654 |
| NGA50 | 5,794,944 | 5,729,014 | 10,816,842 | **6,360,576** | 11,713,900 |
| NGA75 | 1,511,206 | 1,495,174 | 2,701,541 | **1,596,102** | 2,938,187 |
| # misassemblies | 1,356 | 1,299 | 1,602 | **1,278** | 1,515 |
| Complete BUSCOs | **235 (92.2%)** | 234 (91.8%) | 231 (90.6%) | **235 (92.2%)** | 231 (90.6%) |
| Duplicated BUSCOs | 5 (2.0%) | **4 (1.6%)** | 4 (1.6%) | **4 (1.6%)** | 4 (1.6%) |
| Missing BUSCOs | 11 (4.3%) | **10 (3.9%)** | 13 (5.1%) | **10 (3.9%)** | 13 (5.1%) |

**Notes.**
[a] auN is a new metric to measure assembly contiguity *Li (2020)*.

## Validation on real data

We downloaded genome assemblies for which a DLE optical map was available: the NA12878 human genome (*Jain et al., 2018*), *Brassica oleracea* HDEM (PRJEB26621, *Belser et al., 2018*), *Brassica rapa* Z1 (PRJEB26620, *Belser et al., 2018*), *Musa schizocarpa* (PRJEB26661, *Belser et al., 2018*) and *Sorghum bicolor* Tx430 (PRJNA472170, *Deschamps et al., 2018*). The QUAST and BUSCO (*Simão et al. (2015)*, v4.0.5) tools were used respectively to evaluate the number of misassemblies to the GRCh38.p12 human reference genome and the number of conserved genes among eukaryotes. In all cases, we first used the Bionano workflow to scaffold the draft assembly and launched BiSCoT using the files generated by the Bionano tools (Table 1, Tables S3–S6). The output of the Bionano workflow and BiSCoT are scaffolds, but we generated a contig file for each assembly by splitting each scaffold at every position with at least one N.

Concerning the NA12878 genome, we could detect 515 overlapping regions with a mean size of 47kb and representing in total 24.5 Mb of duplicated sequences. Among these 515 regions, 499 were corrected by BiSCoT using either shared labels (113 regions) or a BLAT alignment (386 regions) when no shared labels were found.

Globally, the contig NX and NGAX metrics increased drastically: the contigs NGA50 of NA12878 increased by around 10%, going from 5.8 Mb to 6.3 Mb. The scaffolds NGAX metrics also increased: the scaffolds NGA50 increased from 10.8 Mb in Bionano scaffolds to 11.7Mb in BiSCoT scaffolds. Moreover, the number of Ns decreased marginally and the number of complete eukaryotic genes stayed the same in scaffolds. More importantly, when aligning the assemblies against the reference genome, we could detect a decrease in the number of mis-assemblies going from 1,602 in Bionano scaffolds to 1,515 in BiSCoT scaffolds. The same kind of results were observed in the four plant genomes with a slight

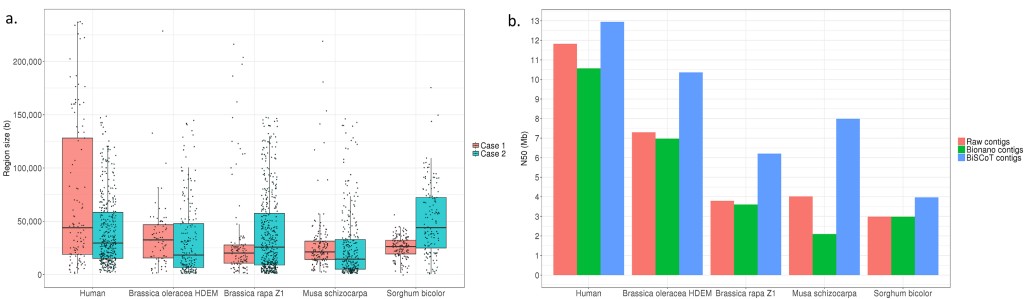

**Figure 2** (A) Distribution of the sizes of overlapping regions in the raw assemblies. Detection was done using either Bionano labels (Case 1) or a BLAT alignment (Case 2). (B) N50 contigs of raw assemblies and assemblies before or after BiSCoT treatment.

decrease in scaffolds NX metrics and number of Ns but an increase in contigs NX metrics (Fig. 2 and Tables S2–S5).

# SUMMARY

Thanks to the advent of long reads and optical maps technologies, it is now possible to obtain high-quality chromosome-scale assemblies. However, the official Bionano scaffolding tool does not always perform optimally when joining two contigs. Indeed, it does not merge two sequences when they share a genomic region, creating artificial gaps in the assembly. We developed BiSCoT, a tool that corrects these problematic regions in a prior Bionano scaffolding and showed that it increased significantly contiguity metrics of the resulting assembly, while preserving its quality.

# ACKNOWLEDGEMENTS

The authors are grateful to the Bionano Genomics staff for technical help and would also like to thank the Whole Human Genome Sequencing Project for providing access to the Nanopore human genome assembly.

## Funding

This work was supported by the Genoscope, the Commissariat à l'Energie Atomique et aux Energies Alternatives (CEA) and France Génomique (ANR-10-INBS-09-08). The funders had no role in study design, data collection and analysis, decision to publish, or preparation of the manuscript.

## Grant Disclosures

The following grant information was disclosed by the authors:
The Genoscope, the Commissariat à l'Energie Atomique et aux Energies Alternatives (CEA).
France Génomique: ANR-10-INBS-09-08.
## Competing Interests

The authors declare there are no competing interests.

## Author Contributions

- Benjamin Istace conceived and designed the experiments, performed the experiments, analyzed the data, prepared figures and/or tables, authored or reviewed drafts of the paper, and approved the final draft.
- Caroline Belser conceived and designed the experiments, performed the experiments, analyzed the data, authored or reviewed drafts of the paper, and approved the final draft.
- Jean-Marc Aury conceived and designed the experiments, analyzed the data, authored or reviewed drafts of the paper, and approved the final draft.

## Data Availability

Data used and code are available at GitHub:

http://www.genoscope.cns.fr/biscot.

## Supplemental Information

Supplemental information for this article can be found online at http://dx.doi.org/10.7717/peerj.10150#supplemental-information.

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
