# Peer review of "BiSCoT: improving large eukaryotic genome assemblies with optical maps"

_PeerJ, doi:10.7717/peerj.10150_

## Round 0.1 · original submission · Major Revisions

Please follow all suggested changes by both reviewers; in particular provide a more extensive evaluation and provide evidence that your proposed method is an improvement over prior work (cf. comments from Reviewer 2).

Reviewer 1 ·

Basic reporting

The manuscript from Istace et al. is written clearly, is easy to follow, and addresses an interesting question. The language, scientific terminology, and structure is sufficient for a manuscript of this kind.

Experimental design

The authors chose publicly available datasets for one human and three plant genomes. I think this is a strong start. The method is explained well with sufficient detail.

Validity of the findings

The examples used in the paper are good and report results for a human genome and three plant genomes.

I think that the authors need to test their methods on more genomes. It would also be useful if there were a positive control experiment. I understand the correction for contigs when they share anchors or regions or one being a subset of another.

Additional comments

I liked the work presented by the authors in this manuscript. I have a few comments:

1. A positive control could help confirm the validity of the BiSCoT code. I am not sure if such an experiment exists.

2. I am surprised that the method developed by Bionano Genomics doesn't do it as is. As I understand, most community uses the software made by the community, which makes me further speculate the magnitude of the issue brought to light by the authors. Is this a common issue when using optical maps for scaffolding?

3. Is there a risk in doing this contig correction analysis on unphased assemblies? If an issue, this would escalate further in plant genomes.

I send my best wishes to the authors.

Reviewer 2 ·

Basic reporting

The manuscript does not discuss prior work, except regarding the Bionano product. There are ”traditional” scaffolders, working with paired-end and mate-pair reads, that handles overlapping contigs. There are also other scaffolders than Bionano’s, why cannot they be used?

I think you are re-defining the contig concept a bit, which made me confused when reading your results. My understanding of contigs is that they are the result of overlapping and merged reads (short or long) from the assembler. Then a scaffolder is further putting contigs together using auxiliary information (such as linked reads or optical maps, sometimes ever RNA evidence). It does not make sense to me to classify a scaffold as a contig just because identical sequence has been identified by the scaffolder. I can understand your thinking and will not insist on my definition (which may be outdated), but I think you should clarify this reclassification of scaffolds into contigs.

Please consider making BiSCoT available on PyPI (pipy.org) so that all a user needs to do is ”pip install BiSCoT”. It is actually pretty easy to set it up and it makes your tool so much more available.

Text issues:
* Change ”long reads sequencing” to ”long read sequencing” (first occurrence in the abstract).
* The key sentence in the abstract, "We developed BiSCoT (Bionano SCaffolding COrrection Tool), a tool that uses informations produced by a pre-existing assembly based on optical maps as input and improves the contiguity and the quality of the generated assembly.” has several deficiencies in my opinion. (1) An assembly does not produce information. It might for example ”provide” or ”contain” information, but you could also rephrase. (2) It is unclear what the actual input is. (3) The sentence does not quite state what BiSCoT does, only that it improves quality. It would be good to be more explicit and explain that is is ”post-processing of Bionano scaffolds” or something like that. (4) I was confused about what ”pre-existing assembly” would mean. One can read that as ”any assembly from a repository”, but you are actually only working with Bionano-scaffolded assemblies. (5) The phrase ”… of the generated assembly” is unclear since it is not obvious that your tool is generating an assembly. (6) Finally, ”information” should not be used in plural.
* The word ”apparition” is not strange in this context. ”The appearance of short read technologies” is better.
* I don’t know what an ”endonuclease nicking site” is.
* ”several folds higher” is not good English, use ”several times higher”.
* Change ”doesn’t” to ”does not”.
* ”contained into” should probably be ”contained in”.
* In ”Case 1”, what is meant by "First,the right-most shared label identifier l is extracted”?
* Please define variables and functions clearly. Reading ”C_k (P_l(C_k))” is just confusing (see ”Case 1”). I suggest rewriting in the style ”The position P_l for a label identifier l” etc.
* What is really meant by ”The position l … on both contigs … is searched”?
* The columns in Table 1 should be justified so that it is easy to read and compare numbers. The left-most column should be left-justified and the data-columns should be justified on decimal point.
* Bold text for best values in the tables (including Supplemental tables) is not consistent. There are several places where there are several numbers that are equal but only some are in bold.
* Capitalisation in the reference list should be corrected for words like ”Nanopore”, ”Promethion”, ”Solanum”, ”BLAST”.

Experimental design

## Code ##
I looked at your code. I would strongly recommend to structure the code by defining functions for any kind of logical block of code you have. Limit the length of a function to one screen page. You have one function, ”main”, and it encompasses the program in its entirety. There is not a single block of code that is easy to test. This makes it hard to build upon your code.

I also note that you have code like
try:

except:
pass

That is a bad practice. If an error occurs, the ”pass” statement simply hides it from the user. There is usually a good reason for an exception to occur, so don’t hide them — handle them.

## Method ##
You start the method section with "Briefly, BiSCoT loads all necessary files into memory”. That is not specific enough for me. I should not need to guess what is considered needed. In the same sentence, you refer to alignments, but at that point it is not clear where the alignments come from.

For Case 2, for contigs that are linked (I presume, it is a bit unclear) but do no share labels, you align contig ends if the estimated distance is at most 1000 nt. Why 1000? Is that a ”safe” distance?

I am deducing that the distance between two contigs is _estimated_ using Equation 1, but I think you should make that clear. How well is that distance estimated?

It is not clear to me that ”Case 3”, merging of contigs sharing labels indicating that one contig _may_ have accidentally been excised from another contig. How can you exclude that the smaller (in your case) contig is not simply a partial duplication? What is the rationale?

Validity of the findings

I find the findings believable, but it is a weakness that there are only four test assemblies.

Additional comments

This manuscript reports a tool to refine scaffolder output from Bionano tools to identify and fix cases of linked contigs having overlap. The method has been tested on four genome assemblies. The code is publicly available and the small scale testing is on public data.

---

## Round 0.2 · accepted · Accept

I apologize for the delays, but COVID-19 continues to gravely impact every academic's schedule.

Reviewer 1 ·

Basic reporting

Thank you addressing reviewer comments. I do not have any further questions, and send my best wishes to the authors.

Experimental design

no comment

Validity of the findings

no comment

Reviewer 2 ·

Basic reporting

Basic reporting looks good.

Experimental design

The experiments are still somewhat limited, but given the scale of data and the type of tool we are looking at, the tests are sufficient to make the point the authors are looking for.

Validity of the findings

Good.

Additional comments

I am pleased with the improved manuscript.